# Molecular Mechanisms of Functional Adrenocortical Adenoma and Carcinoma: Genetic Characterization and Intracellular Signaling Pathway

**DOI:** 10.3390/biomedicines9080892

**Published:** 2021-07-26

**Authors:** Hiroki Shimada, Yuto Yamazaki, Akira Sugawara, Hironobu Sasano, Yasuhiro Nakamura

**Affiliations:** 1Division of Pathology, Faculty of Medicine, Tohoku Medical and Pharmaceutical University, 1-15-1 Fukumuro, Miyagino-ku, Sendai 983-8536, Miyagi, Japan; shimadahis@tohoku-mpu.ac.jp; 2Department of Pathology, Tohoku University Graduate School of Medicine, 2-1 Seiryo-machi, Aoba-ku, Sendai 980-8575, Miyagi, Japan; y.yamazaki@patholo2.med.tohoku.ac.jp (Y.Y.); hsasano@patholo2.med.tohoku.ac.jp (H.S.); 3Department of Molecular Endocrinology, Tohoku University Graduate School of Medicine, 2-1 Seiryo-machi, Aoba-ku, Sendai 980-8575, Miyagi, Japan; akiras2i@med.tohoku.ac.jp

**Keywords:** aldosterone-producing adenoma (APA), cortisol-producing adenoma (CPA), adrenocortical carcinoma (ACC), gene mutation, transcription factors

## Abstract

The adrenal cortex produces steroid hormones as adrenocortical hormones in the body, secreting mineralocorticoids, glucocorticoids, and adrenal androgens, which are all considered essential for life. Adrenocortical tumors harbor divergent hormonal activity, frequently with steroid excess, and disrupt homeostasis of the body. Aldosterone-producing adenomas (APAs) cause primary aldosteronism (PA), and cortisol-producing adenomas (CPAs) are the primary cause of Cushing’s syndrome. In addition, adrenocortical carcinoma (ACC) is a highly malignant cancer harboring poor prognosis. Various genetic abnormalities have been reported, which are associated with possible pathogenesis by the alteration of intracellular signaling and activation of transcription factors. In particular, somatic mutations in APAs have been detected in genes encoding membrane proteins, especially ion channels, resulting in hypersecretion of aldosterone due to activation of intracellular calcium signaling. In addition, somatic mutations have been detected in those encoding cAMP-PKA signaling-related factors, resulting in hypersecretion of cortisol due to its driven status in CPAs. In ACC, mutations in tumor suppressor genes and Wnt-β-catenin signaling-related factors have been implicated in its pathogenesis. In this article, we review recent findings on the genetic characteristics and regulation of intracellular signaling and transcription factors in individual tumors.

## 1. Introduction

Adrenocortical tumors are broadly classified into adenomas and carcinomas based on their potential biological behavior. In addition, adrenocortical adenomas are further subdivided into functional adenomas that secrete excessive steroid hormones and non-functional ones which do not. In this review article, we will review the findings of recently reported studies on genetic alterations and their regulation of intracellular signaling in aldosterone-producing adenoma (APA) as a cause of primary aldosteronism (PA) and cortisol-producing adenoma (CPA) as a cause of Cushing’s syndrome, subclinical Cushing’s syndrome, and adrenocortical carcinoma (ACC). APA is an adenoma producing excessive aldosterone autonomously, and somatic mutations of ion channels located at the cell membrane have been frequently reported, resulting in alteration of calcium signaling and its downstream transcription factors [1,2,3,4,5]. CPA is an autonomous cortisol-producing adenoma in which somatic mutations in genes encoding those involved in intracellular cAMP-PKA signaling [6,7,8,9,10,11,12,13,14,15] have been reported to be associated with their pathogenesis. ACC is a highly malignant cancer originating from the adrenal cortex, and mutations in tumor suppressor genes [16,17,18], those involved in Wnt-β-catenin signaling [19,20,21], and chromatin remodeling factors [19,20] have been reported to contribute to its pathogenesis. In addition, transcription factors have been also reported to regulate the expression of downstream genes by binding to other chromatin-related proteins and epigenomic regulators to form transcription factor complexes. We will also review the relationship between gene mutations and their regulation of transcription factors, as well as the association of transcription factor complex formation in APA, CPA, and ACC.

## 2. The Pathogenesis and Molecular Mechanisms of Aldosterone Overproduction in Aldosterone-Producing Adenoma (APA)

Somatic mutations in *KCNJ5, ATP1A1, ATP2B3*, *CACNA1D*, *CACNA1H*, *CLCN2*, and *CTNNB1* genes have been frequently detected in APA [1,2,3,22,23,24,25]. *KCNJ5* encodes a rectifying potassium ion channel (Kir 3.4) that regulates resting cellular membrane potential. The genetic variants and hot spots of somatic mutations in *KCNJ5* are well characterized, including L168R, G151R, T158A, G151E, I157del, T152C, and E154Q [26,27]. All of these somatic mutations can occur at ion-selective sites and cause loss of ion selectivity, leading to persistent depolarization due to sodium ions influx into the cell. The depolarization of cells enhances intracellular calcium signaling, which induces the expression of CYP11B2, one of the rate-limiting enzymes in aldosterone synthesis (Figure 1). CYP11B2 expression is mediated by orphan nuclear receptors such as *NR4A1* (NGFIB), *NR4A2* (NURR1), and *NR4A3* (NOR1) [28,29] and the activation of calcium signaling via CaM-CaMK [27,30]. It has also been reported that overexpression of mutant *KCNJ5* in cell line experiments increased intracellular calcium ion concentration due to depolarization, enhanced expression of NURR1, and induced expression of CYP11B2 [31]. These results suggest that the hypersecretion of aldosterone caused by the *KCNJ5* mutation is mediated by the activation of NR4A family orphan receptors via calcium signaling through the increase in intracellular calcium ion concentration, which in turn induces the expression of aldosterone synthesis genes, including *CYP11B2* (Figure 2).

*ATP1A1* encodes a sodium-potassium cotransporter (sodium/potassium-transporting ATPase subunit alpha-1), a protein whose essential function is to export sodium ions out of the cell and transport potassium ions into the cells. Genetic variants of somatic mutations in *ATP1A1* include L104R, del100_104, V332G, and G99R [2,22,33]. *ATP1A1* L104R loses ion selectivity and allows hydrogen ions to enter the cell, resulting in an increase in aldosterone production via cell depolarization or intracellular acidification [34]. *ATP1A1* del100_104 is another genetic variant that causes loss of ion selectivity, and it has been reported that the influx of sodium ions, due to the loss of ion selectivity, causes cell depolarization (Figure 2).

The ATP2B3 gene [2,33] encodes a calcium transporter that consumes ATP and removes calcium from cells. L425_V426del and V426_V427del are the major genetic variants of somatic mutations detected in ATP2B3 genes, which cause the deletion of amino acids in the calcium ion-exporting subunit, resulting in the loss of intracellular calcium ion export. As a result, the increased intracellular calcium ion concentration induces depolarization and activation of calcium signaling, which promotes the expression of genes involved in aldosterone synthesis (Figure 2).

*CACNA1D* encodes an L-type calcium ion channel (CaV1.3), and somatic mutations such as G403R, S652L, F747L, and R990H have been reported [3,35,36,37], while G403R, S652L, F747L, and R900H mutations change calcium ion gating, and these phenotypes are a gain-of-function in calcium channels. *CACNA1H* encodes a T-type calcium ion channel (CaV3.2), and somatic mutations such as I1430T and T4289C have been reported [1]. Mutations in *CACNA1H* lead to gain-of-function, which is a decrease in the threshold of the potential for calcium ion influx into the cell, increasing intracellular calcium ion concentration, subsequently triggering depolarization and promoting aldosterone secretion (Figure 2).

*CLCN2* is a gene coding for chloride ion channels, and somatic mutations such as G24D, 64-2-74del, and R172G have been reported [4,5,25,38]. *CLCN2* has the ability to efflux chloride ions from cells. The *CLCN2* gene mutation increases the ability of *CLCN2* to efflux chloride ions out of the cell and induces depolarization by disrupting the ion gradient in and out of the cell (Figure 2).

*CTNNB1* is a transcription factor involved in Wnt-β-catenin signaling and encodes β-catenin. Somatic mutations in *CTNNB1* are detected not only in adenomas but also in ACC [18]. β-catenin is constantly degraded by phosphorylation. Inhibition of β-catenin phosphorylation prevents its degradation, resulting in its migration and activity as a transcription factor. Somatic mutations such as S33C, S45F, and S45P, which correspond to the phosphorylation sites of β-catenin, are the major variants [39,40]. Recently, it has been suggested that β-catenin induces the expression of NR4A family proteins and is involved in *CYP11B2* gene expression. This suggests that the increased transcriptional activity of β-catenin may also contribute to aldosterone oversecretion (Figure 3).

The expression of *CYP11B2* and *HSD3B* genes is upregulated in APA, and calcium signaling plays a major role in *CYP11B2* gene expression. In addition to the driver gene somatic mutations mentioned above, various molecules that promote aldosterone biosynthesis and could contribute to the possible pathogenesis of PA have also been reported. There are also reports on the correlation between PA and factors that regulate intracellular calcium ion concentration [41]. TASK is a potassium channel whose function is attenuated by acidic extracellular conditions and the activation of Gq-coupled receptors. *TASK* is also highly expressed in the adrenal cortex, and *TASK1* and *Task1/Task3*-deficient mice are particularly susceptible to depolarization of the adrenal cortex. *TASK1* and *Task1/Task3* deficient mice have been reported as in vivo models of PA, based on their findings of induction of depolarization in the adrenal cortex [42]. PCP4 is a protein that promotes the binding and dissociation of CaM and calcium ions, which are components of calcium signaling. It has been reported that PCP4 is highly expressed in APA, and because PCP4 KD reduces the gene expression of *CYP11B2* [43], it may be a factor that regulates the gene expression of *CYP11B2*. Calneuron1 is a calcium-binding protein that transports cytosolic calcium ions to the endoplasmic reticulum (ER), suggesting its roles in promoting Ca ion accumulation in the ER. Calneuron1 is also reported to be highly expressed in APA [44], suggesting that it is also an important factor in APA (Figure 4). 

*HSD3B2* was more highly expressed in the adrenal cortex than *HSD3B1.* However, *HSD3B2* expression was not significantly increased in normal tissues and APA. In contrast, *HSD3B1* is significantly upregulated in APA tissues compared to the normal adrenal cortex, suggesting that *HSD3B1* is an important factor in APA [45]. Gene expression of *HSD3B1* is also dependent on calcium signaling and increases in response to the transcriptional activity of the NR4A family of transcription factors, which are considered downstream transcription factors of calcium signaling. Therefore, the above calcium signal regulators may be involved in the gene expression of *CYP11B2* as well as *HSD3B1*.

The promoter region of the *CYP11B2* gene contains transcription factor-binding sequences, such as the NGFI-B response element (NBRE), Ad4, Ad5, and Ad1. The NR4A family is an orphan nuclear receptor for which no ligand has been identified, and its expression levels and intracellular signaling may contribute to the expression of downstream genes [28,46,47]. The NR4A family is known to regulate the expression of downstream genes in response to their expression levels and intracellular signals. The exposure of H295R cells, a human ACC-derived cell line, to hyperglycemic conditions was reported to induce the expression of NURR1, a member of the NR4A family of transcription factors, and *CYP11B2* [48]. It is known that transcription factors interact with transcription factor complexes, a group of proteins that alter the epigenomic status of chromatin to regulate the transcription of target genes. NGFIB interacts with RXR, a heterodimer partner, as well as with p300 [49], SRC-1 [50], and other histone modification factors and regulates the expression of downstream genes. NURR1 is known to interact with RXR and SRC-1 [51] as well as NGFIB. However, there are few reports on the analysis of transcriptional activation complexes in the adrenal gland for both NGFIB and NURR1, and the details of transcription factor complexes that may affect gene expression, such as *CYP11B2*, still remain unknown. In 2018, we identified poly (ADP-ribose) polymerase 1 (PARP1) as a transcription factor complex of NURR1 in H295R cells [52] (Figure 5). This is an in vitro study, but it could contribute to our understanding of the unknown transcription factor complex of NURR1. SF-1 has been identified as a transcription factor that binds to the Ad4 region, also known as *Ad4BP*, and is known to be essential for adrenal and gonadal development. SF-1 expression is also reported to be upregulated in APA [53], suggesting that it affects aldosterone synthesis in APA. SF-1 has been reported to be involved in steroid synthesis. Recently, it was reported that SF-1 regulates not only the expression of steroid synthase but also the expression of glycolytic enzymes [54] and cholesterol synthase [55]. COUP-TF is a transcription factor that binds to the Ad5 region and represses the transcriptional activity of CYP11B2 in the adrenal cortex. Recently, it has been reported that COUP-TF contributes to the repression of transcriptional activity by interacting with Ubc9, a SUMO-transferase [56]. 

On the other hand, there are few reports on the expression level of APA, and the relationship with steroidogenesis awaits further investigation. The promoter regions of the *HSD3B1* and *HSD3B2* genes also contain NBRE, Ad4, Ad5, and Ad1 as transcription factor binding sequences [45,57]. The promoter regions of the *HSD3B1* and *HSD3B2* genes also contain NBREs, suggesting that transcription is regulated by the NR4A family of transcription factors.

In addition, new therapeutic agents have recently been investigated. In particular, macrolide antibiotics have been suggested to inhibit aldosterone secretion by suppressing the function of *KCNJ5* mutations [58]. It has been suggested that macrolide antibiotics inhibit cell-autonomous depolarization by binding to mutated *KCNJ5* and preventing the influx of sodium ions into the cell. Similarly, for treating APA, it is thought that downregulating the expression of *HSD3B1*, one of the genes in the aldosterone synthesis system, and CYP11B2, the rate-limiting enzyme in aldosterone synthesis, suppresses excessive aldosterone secretion. *HSD3B1* gene expression is induced by orphan nuclear receptors such as NGFIB, NURR1, and NOR1 [45,57]. NGFIB, NURR1, NOR1, COUP-TF, SF-1, and cAMP-response element-binding protein (CREB) are transcription factors that induce CYP11B2 gene expression [28,47]. 

Among them, NURR1 is a nuclear receptor whose expression increases when the intracellular calcium concentration transcriptional activity increases. NURR1 also interacts with PARP1 to regulate the transcriptional activity of target genes [52]. In vitro, it has been reported that inhibitors of PARP1 suppress aldosterone secretion by suppressing the expression of *HSD3B1* and *CYP11B2* [52]. In addition, bortezomib was reported to decrease *CYP11B2* gene expression by altering the epigenomic state upstream of the *CYP11B2* gene [59]. In addition, the drug-induced activation of other nuclear receptors, such as PPARγ and RXR with ligands, was reported to decrease CYP11B2 expression via NR4A family transcription factors such as NURR1 [60,61]. Although there are still many unanswered questions on potential nuclear receptor crosstalk, this is an interesting finding regarding the role of nuclear receptors in the adrenal cortex. There are also reports of compounds that inhibit aldosterone secretion by directly inhibiting the enzymatic activity of CYP11B2. Reports on the treatment of APA have focused on the development of drugs targeting the driver genes involved in aldosterone excess in APA, such as *KCNJ5*, as well as on the gene expression and enzyme activity of *CYP11B2* and other enzymes in the aldosterone synthesis system.

## 3. The Pathogenesis and Molecular Mechanisms of Cortisol Production in Cortisol-Producing Adenoma (CPA) 

CPAs, which secrete cortisol autonomously, cause Cushing’s syndrome and subclinical Cushing’s syndrome. Recently, CPA has been reported to be caused by somatic mutations of cAMP-PKA signaling factors, such as *PRKACA*, *PRKACB*, *PRKAR1A*, and *PRKAR1B* mutations [6,14,62], as well as genetic mutations in cAMP-degrading enzymes such as PDE8B and PDE11A [11,12] (Figure 6). In addition, GNAS mutations that enhance MC2R function are also causative mutations in CPA [63].

*PRKACA* and *PRKACB* both constitute the enzymatically active subunits of PKA, and their mutations cause direct changes in the enzymatic activity of PKA [6,14,15,62]. *PRKACA* mutations were also reported in L206R and L199_C200insW, while *PRKACB* mutations were reported in S54L. Both genetic mutations also inhibit the recruitment of regulatory subunits and increase the kinase activity of PKA, which results in increased gene expression of *CYP11B1* and increased secretion of cortisol. Recently, gene expression profiling of PRKACA-mutated CPA has been performed by RNA-seq analysis, and it is expected to reveal the whole picture of gene expression in PRKACA-mutated CPA [64].

*PRKAR1A* encodes the regulatory subunit 1α of PKA, and *PRKAR1B* encodes the regulatory subunit 1β of PKA, which has two subunits: a catalytic subunit that has enzymatic activity and a regulatory subunit that controls enzymatic activity. Loss-of-function mutations in the regulatory subunit increase the enzymatic activity of PKA and activate PKA signaling. The activation of PKA signaling in the adrenal cortex induces the expression of *CYP11B1*, which promotes cortisol secretion, suggesting that the loss of function of the regulatory subunit of PKA is associated with the hypersecretion of cortisol. p.I40V, p.A67V, p.A300T, and other mutations in *PRKAR1B* have been reported [7]. p.A300T and p.A67V are reported mutations that decrease the activity of PKA [65], which is contradictory considering that PKA activation increases the gene expression of *CYP11B1*, but further studies are required for clarification. 

In contrast, phosphodiesterase (PDE) family proteins, to which PDE8B and PDE11A belong, function as cAMP-degrading enzymes. When intracellular cAMP is degraded by PDE family proteins, PKA decreases its enzymatic activity due to a decrease in intracellular cAMP concentration. Mutation of PDE family proteins causes the loss of enzymatic activity, which leads to the loss of intracellular cAMP degradation, an increase in intracellular cAMP concentration, and an increase in PKA activity. It has been reported that PDE8B is predominantly expressed in the adrenal glands and other steroid hormone-secreting organs. H305P decreases the enzymatic activity of PDE8B and increases the concentration of intracellular cAMP [9,11]. PDE11A, D609N, and M878V are mutations that decrease the enzymatic activity of PDE11A [9,12], similar to PDE8B mutations, and increase the concentration of intracellular cAMP. Due to the increased cAMP, PKA activity is subsequently increased, resulting in increased expression of *CYP11B1*, leading to hypersecretion of cortisol.

Mutations in the guanine nucleotide-binding protein subunit alpha (*GNAS*) gene, a G-protein alpha-subunit associated with MC2R, have also been reported in CPA. *GNAS* mutations activate cAMP-PKA signaling, which in turn enhances the expression of the downstream *CYP11B1* gene [63]. The promoter region of the *CYP11B1* gene contains transcription factor binding sequences such as Ad4, Ad5, and Ad1 [66]. Ad4 binds SF-1, while Ad5 binds orphan nuclear receptors such as those in the NR4A family and COUP-TF. In addition, Ad1 binds cAMP-PKA-responsive transcription factors such as CREB and CREM. CREB and CREM are thought to be important in CPA, where genetic mutations in cAMP-PKA signaling factors are common. Previous in vitro studies have shown that the Ad1 region plays a central role in transcriptional activation when cAMP-PKA signaling is activated, and CREB and CREM are considered the most important transcription factors in CPA. The CREB transcription factor-activating complex is composed mainly of HATs, such as p300 [67,68] and CREB-binding protein (CBP) [69] (Figure 7). However, there are no reports on the analysis of the CREB transcription factor complex in the adrenal gland, which is an important issue to be addressed in the future. The interaction of CREB with β-catenin and glucocorticoid receptors (GR) has been reported in other organs [70], which may provide interesting results.

## 4. The Pathogenesis and Molecular Mechanisms in Adrenocortical Carcinoma (ACC)

ACC is a malignant tumor arising from the adrenal cortex. The frequency of ACC is relatively low (approximately 1–2 per million), but its prognosis is poor or dismal due to its rapid clinical growth and progression. There are two types of ACCs: functional tumors that are based on the presence of hormone excess and nonfunctional tumors that do not secrete hormones.

Genetic mutations in adrenocortical malignancies have been detected in tumor suppressor genes such as *TP53* and *RB1* [17,71,72], mutations in Wnt-β-catenin system genes such as *ZNRF3* and *CTNNB1* [17,21,71], mutations in cell cycle-related genes such as *CDKN2A* [17], and mutations in epigenomic and chromatin remodeling regulators such as *DAXX*, *MED12*, and *M**EN1* [17,71]. In addition, *IGF**2* (insulin-like growth factor 2), *PRKAR1A*, *RPL22* (ribosomal protein L22), *CCNE1* (cyclin E1), *CDK4* (cyclin dependent kinase 4), *TERT* (telomerase reverse transcriptase), and *TERF2* (telomeric repeat binding factor 2) have also been reported as genetic mutations in ACC [20].

Mutations in tumor suppressor genes are also frequently reported in cancers of other organs, and they play important roles in the pathogenesis of ACC.

*TP53* mutations in ACC are nonsense or frameshift mutations that cause a loss of *TP53* function [20]. *TP53* gene is also known to be the causative gene of Li-Fraumeni syndrome. *TP53* encodes p53 protein. The p53 protein is constantly degraded and accumulates in the cells when it detects DNA damage and suppresses cell growth [73]. The deletion of p53 leads to uncontrolled cell proliferation, which in turn leads to the growth of carcinoma cells. *RB1* is a tumor suppressor gene detected in retinoblastoma, and RB protein binds to the transcription factor E2F and inhibits its transcriptional activity, thereby suppressing cell growth [74]. In ACC, RB has been reported to be functionally suppressive, suggesting that the RB-mediated cell growth suppression mechanism is disrupted [17,18].

Mutations in the Wnt-β-catenin system, an important intracellular signal for cell proliferation, are also important. β-catenin is a transcription factor that is constantly degraded, and its degradation mechanism is regulated by the phosphorylation of β-catenin [75]. Wnt proteins bind to the Frizzled receptor and couple with the LRP receptor, inactivating GSK3B, which is a phosphatase of β-catenin, allowing β-catenin protein to be spared from degradation [75,76]. As a result, β-catenin accumulates in the cells and contributes to the expression of downstream genes. Of note, the *CTNNB1* mutation was frequently detected in ACC, suggesting its possible contribution to tumor growth or progression. β-catenin is a transcription factor, and its activation depends not only on the accumulation of β-catenin in the nucleus by post-translational modification but also on the effect of transcription factor complexes. Therefore, it is expected that the epigenomic factors that construct the transcription factor complex will be elucidated in the future, in order to be able to develop drugs that specifically target it.

*ZNRF3* is a cell surface ubiquitin ligase that regulates Wnt-β-catenin signaling by ubiquitinating and degrading the Wnt protein receptors Frizzled and LRP6 [77]. When *ZNRF3* loses its activity due to mutation, Wnt-β-catenin signaling is not regulated and β-catenin activity is increased [17,71] (Figure 8). 

Additionally, mutations in epigenetic regulators may increase the transcriptional activity of β-catenin, which may be involved in tumorigenesis. 

*MED12* (mediator of RNA polymerase II transcription subunit 12 homolog) is a known factor that interacts with RNA polymerase. *MED12* is involved in the activity of β-catenin, and its mutation has been reported to increase the activity of Akt and decrease the activity of GSK3B, an upstream regulator of β-catenin, resulting in increased transcriptional activity of β-catenin [78,79,80]. This suggests that *MED12* mutations contribute to the growth of malignant tumors [17,79,80]. However, the details of molecular regulation between mutations and growth, as well as changes in chromatin status and cancer caused by *MED12* mutations, remain unclear, and future studies are expected (Figure 8).

*MEN1* encodes Menin, is a tumor suppressor gene and a causative gene of multiple endocrine neoplasia type 1 (*MEN1*). *MEN1* acts as a transcriptional repressor in the epigenome, forming protein–protein interactions with various transcription factors and repressing their transcriptional activity. It has been reported that *MEN1* also binds to β-catenin and affects its transcriptional activity, while β-catenin has been reported to activate transcription [81,82]. Mutations in *MEN1* have been implicated in the pathogenesis of *MEN1*, and Menin promotes the expression of p27 mutations in *MEN1*, which have been implicated in the pathogenesis of *MEN1*. However, the mechanism by which Menin promotes p27 expression and inhibits cell proliferation is thought to be disrupted by genetic mutations [82]. The same is thought to be true in adrenal cancers.

*DAXX*, a chromatin remodeling factor, affects histone H3.3 and promotes telomere elongation. It also interacts with β-catenin and TCF4 to increase its transcriptional activity [83]. Mutations of *DAXX* in ACC are often reported to be deletion type mutations, and their relevance to proliferation is currently unclear.

*IGF2* is a key factor in adrenal development, and increased gene expression has been reported in ACC [84]. IGF2 has been reported to proliferate H295R cells [85], suggesting that it is an important factor in the proliferation of ACC. Recently, novel gene mutations have been investigated in *RPL22*, *CCNE1*, *CDK4*, *TERT*, and *TERF2* in ACC using RNA sequencing [20]. *CCNE1* and *CDK4* are cell cycle regulating genes, which control cell proliferation, while *TERT* and *TERF2* are telomere-related genes, which regulate cell senescence and immortality.

Additionally, ACC has been investigated using next-generation sequencing, and mutations in genes such as *LRIG1*, *ZFPM1*, *CRIPAK*, *GARS*, and *ZNF517* have been discovered [16]. Although the correlation between adrenocortical cell tumorigenesis and cancer malignancy remains unclear, it is expected to be a topic for future research on genetic mutations in ACC.

As mentioned above, β-catenin, which plays a critical role in adrenal carcinoma, binds to the transcription factors TCF or LEF to form transcription factor complexes. p300 [86], CBP [87], and other HATs, and MLL1/2 [88] belonging to the lysine-specific methyltransferase (KMT) family of proteins, are representative components of β-catenin transcription factor complexes. However, the β-catenin transcription complex in the adrenal gland has not been studied in detail, and the analysis of organ-specific transcription factor complexes has not progressed.

## 5. Conclusions

In APA, the cause of PA, the upregulation of aldosterone synthesis genes by calcium signaling is a key cascade that enhances aldosterone secretion by activating transcription factors, mainly the NR4A family. In CPA, the activation of cAMP-responsive transcription factors such as CREB increases cortisol secretion by upregulating the gene expression of *CYP11B1*. The Wnt-β-catenin system is important in ACC, and the suppression of β-catenin transcriptional activity is important in future therapeutic development. Mutations of β-catenin have been reported in adenomas, but β-catenin is mainly involved in tumorigenesis, and in adenomas, it seems to promote steroid production as a secondary effect by affecting the NR4A family and CREB. In addition, the presence of transcription factor complexes is important for the activation of transcription factors, and future studies focusing on transcription factors and transcription factor complexes are warranted in APA, CPA, and ACC.

## Figures and Tables

**Figure 1 biomedicines-09-00892-f001:**
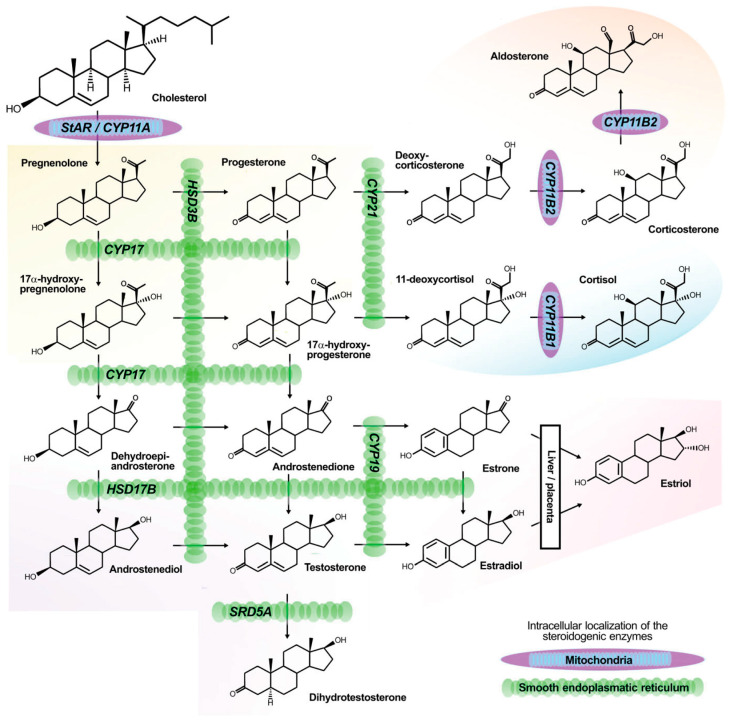
The steroidogenic pathways and their involved enzymes and products [32].

**Figure 2 biomedicines-09-00892-f002:**
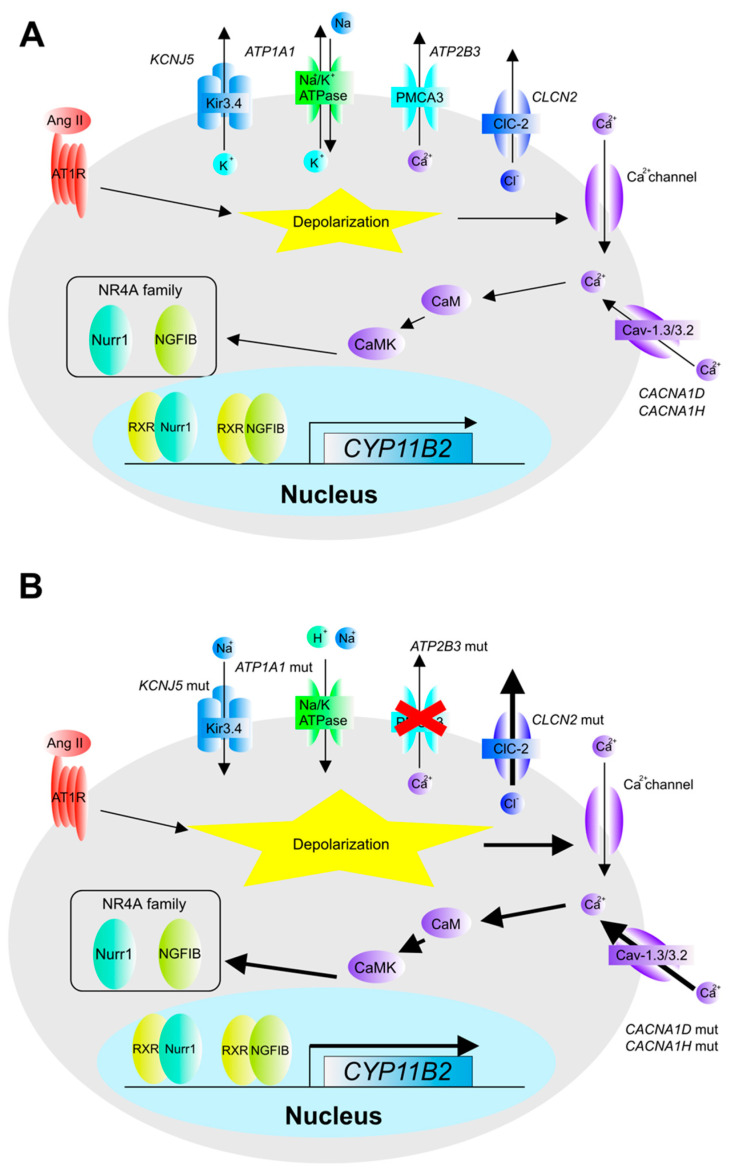
Gene mutations of membrane protein in APA. Depolarization of the plasma membrane is the common consequence in this mechanism. (**A**) Normal adrenal membrane proteins. (**B**) Gene mutation in adrenal membrane proteins. *K**CNJ5* mutation causes inflow of sodium ion; *ATP1A1* mutation causes hydrogen and sodium ion leak; *ATP2B3* mutation causes impaired calcium ion release; *CLCN2* mutation causes impaired chloride ion release; and calcium ion influx is due to mutations in *CACNA1D* and *CACNA1H*.

**Figure 3 biomedicines-09-00892-f003:**
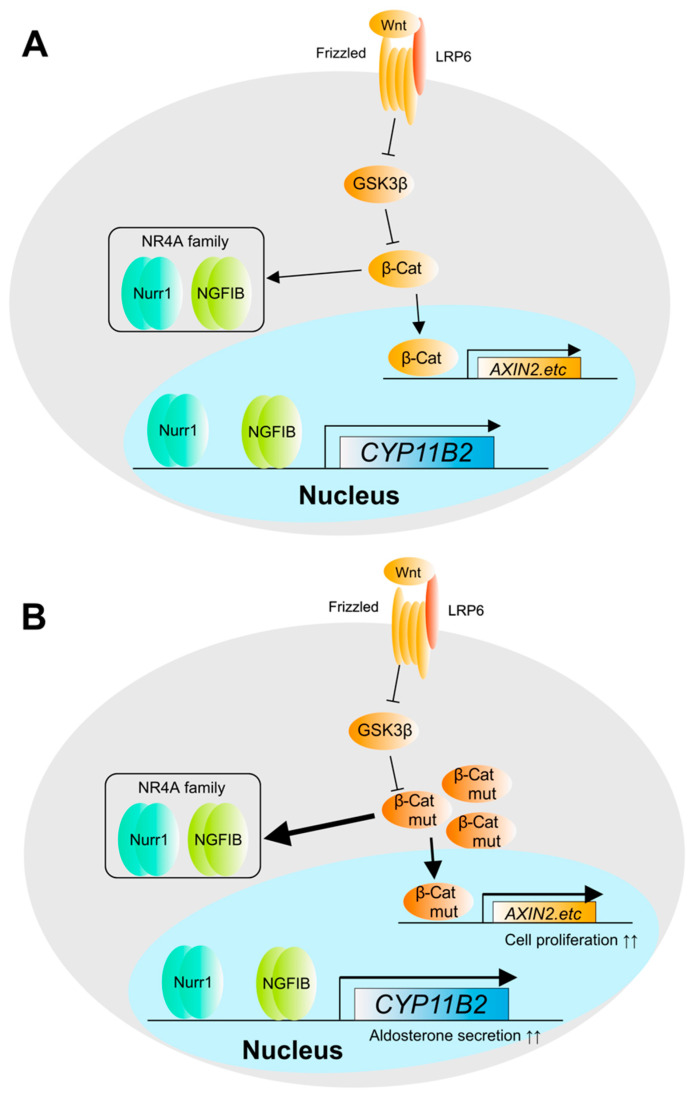
Mutation of *CTNNB1* gene in APA. Mutations in β-catenin have been reported to activate the pathway independent from cell membrane depolarization in APA. (**A**) Normal adrenal β-catenin pathway. (**B**) Gene mutation in adrenal β-catenin pathway. β-catenin accumulates intracellularly when Wnt binds to Frizzled and induces the expression of downstream genes. β-catenin mutations cause Wnt-independent accumulation of β-catenin without degradation. Mutations in β-catenin cause it to accumulate in a Wnt-independent manner. As a result, it induces cell proliferation by activating downstream genes including *AXIN2* and other cell proliferation-related genes. *AXIN2* is β-catenin regulator via GSK3B. On the other hand, it has been suggested that β-catenin induces the expression of NR4A family proteins and affects *CYP11B2* gene expression.

**Figure 4 biomedicines-09-00892-f004:**
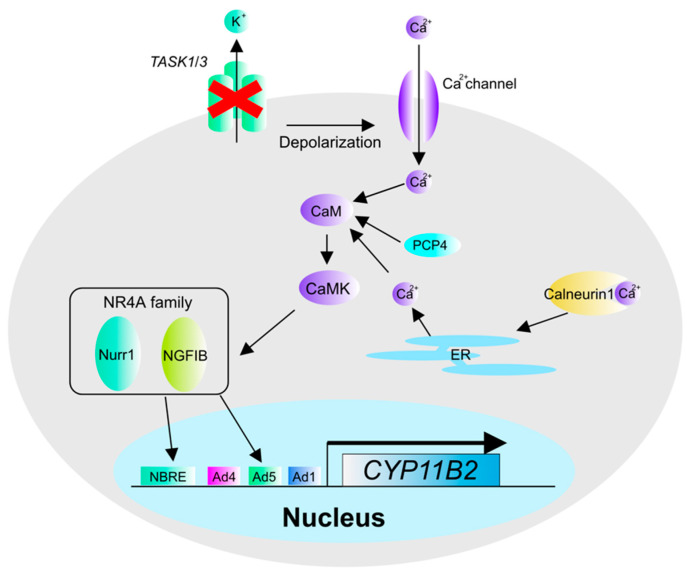
Calcium signaling-related factors in APA. Calcium signaling-related factors that have been implicated in the pathogenesis of APA include the potassium channels TASK, PCP4, and Calneurin1. TASK is functionally defective due to genetic mutations that cause depolarization. PCP4 is a factor that activates CaM function. Calneurin1 is a protein that binds intracellular calcium ions and transports them to the ER, where they enhance calcium signaling during depolarization.

**Figure 5 biomedicines-09-00892-f005:**
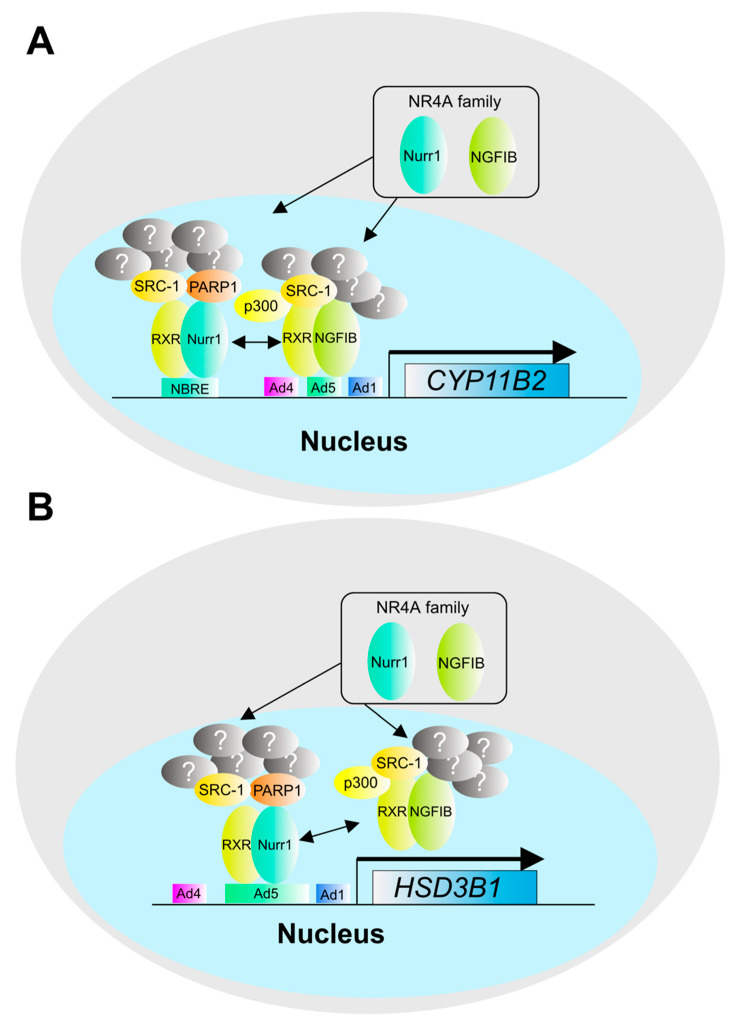
Transcriptional regulation of steroidogenic genes by the NR4A family. The promoter regions of *CYP11B2* and *HSD3B1* contain NR4A family binding sequences such as NGFIB and Nurr1, which regulate gene expression of *CYP11B2* and *HSD3B1* by forming a transcription factor complex. (**A**) Gene expression of *CYP11B2* by NR4A family. (**B**) Gene expression of *HSD3B1* by NR4A family. Although p300, SRC-1, and PARP1 have been reported as components of the complex, other components are still unknown.

**Figure 6 biomedicines-09-00892-f006:**
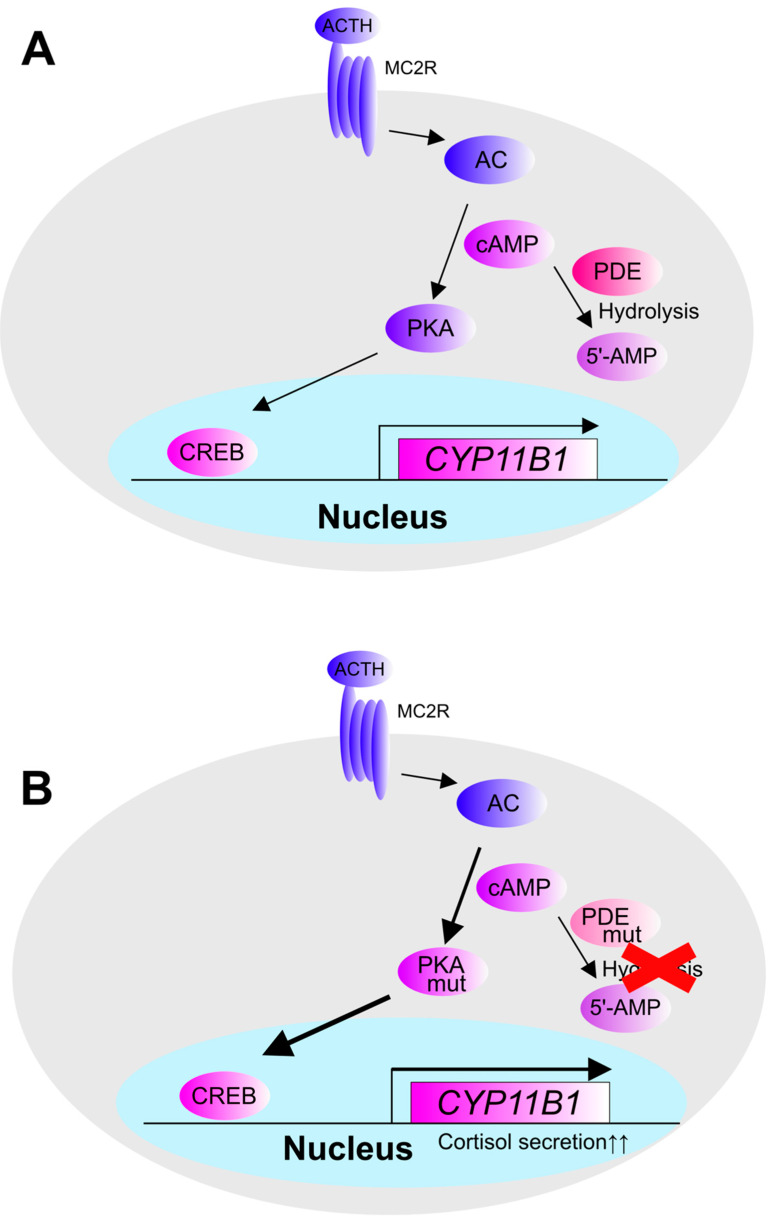
Genetic mutation in CPA. Cortisol is secreted by inducing the gene expression of CYP11B1, and ACTH activates cAMP-PKA signaling by binding to MC2R of ZF cells, inducing the gene expression of *CYP11B1*. (**A**) Normal adrenal cAMP-PKA pathway. (**B**) Gene mutation in adrenal cAMP-PKA pathway. In CPA, mutations in *PRKAR1B*, *PRKACA*, and *PRKACB*, which are subunit genes of PKA, and mutations in *PDE8B* and *PDE11A,* which are enzymes that degrade cAMP, have been reported.

**Figure 7 biomedicines-09-00892-f007:**
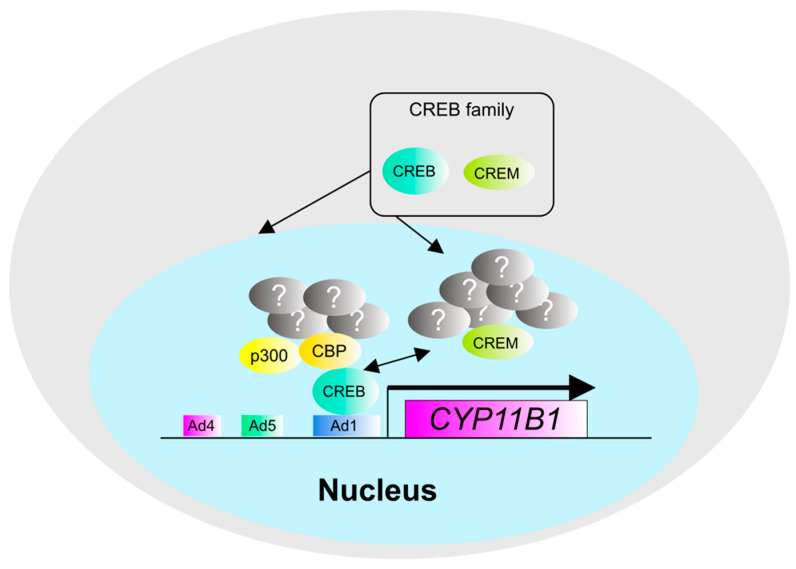
Transcriptional regulation of the *CYP11B1* gene by the CREB family. The CREB family, including CREB and CREM, is involved in the regulation of gene expression of *CYP11B1*, a cortisol synthase. CREB has been identified as a component of the transcription factor complex with p300 and CBP, but the other components are still unknown.

**Figure 8 biomedicines-09-00892-f008:**
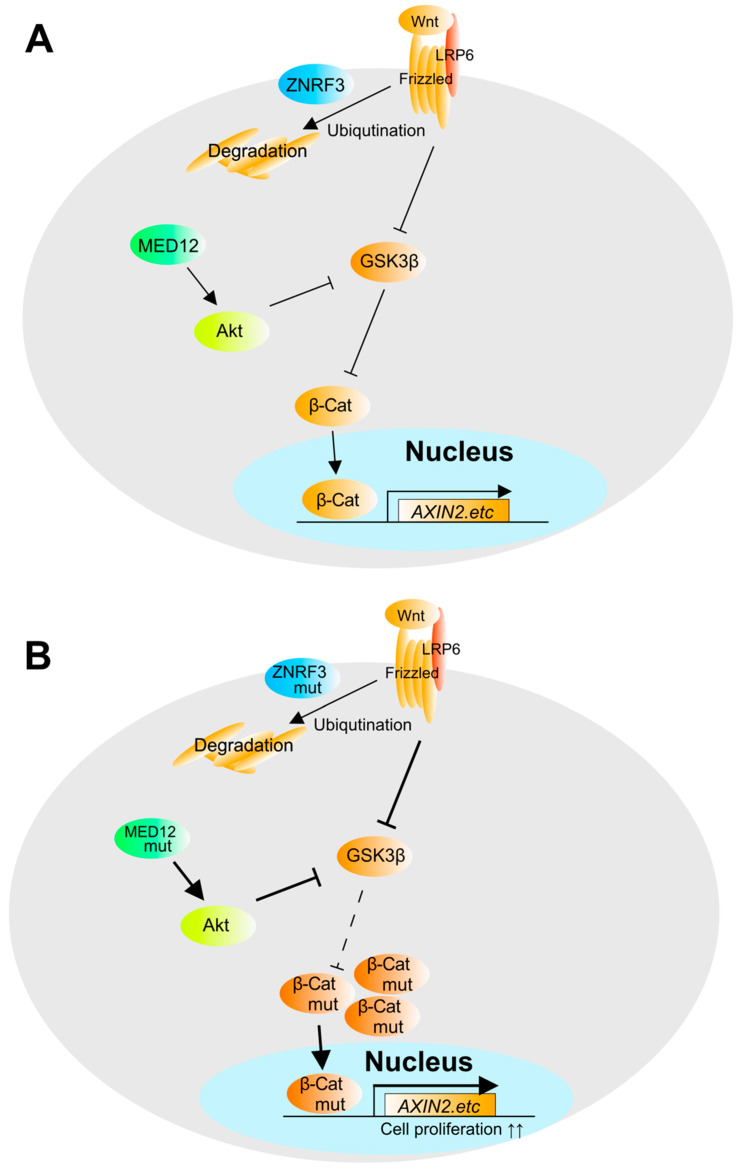
Genetic mutations in adrenocortical carcinoma. In adrenocortical carcinoma, mutations in the β-catenin system have been reported. (**A**) β-catenin pathway in non-ACC. (**B**) β-catenin pathway in ACC. *AXIN2* is β-catenin regulator via GSK3B. *ZNRF3* is a ubiquitin ligase that regulates the degradation of Frizzled and LRP6. Mutation of *MED12* enhances the activation of Akt and further enhances the transcriptional activity of β-catenin.

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
