# Peer review of "Molecular Mechanisms of Functional Adrenocortical Adenoma and Carcinoma: Genetic Characterization and Intracellular Signaling Pathway"

_biomedicines, 2021, doi:10.3390/biomedicines9080892_

Round 1

Reviewer 1 Report

Hiroki Shimada  and co-workers,

in their review address the issue of the relationship between gene mutations and their regulation of transcription factors, as well as the association of transcription factor complex formation in aldosterone-producing adenoma (APA), cortisol-producing adenoma (CPA) and adrenocortical carcinoma (ACC).

This paper is conceived through an initial introduction in which are described the general characteristics of the adrenocortical tumors classified into adenomas (APA and CPA) and carcinomas (ACC); a second part addresses the pathogenesis and molecular mechanisms of aldosterone overproduction in APA, a third part the pathogenesis and molecular mechanisms of cortisol production in CPA and a fourth part the pathogenesis and molecular mechanisms in ACC.  In conclusion, the authors highlight the main somatic mutations that have been discovered in genes coding for various proteins and resulting in aldosterone or cortisol hypersecretion in APA and CPA, respectively, or in ACC pathogenesis.

The review is well written and updated in dealing with the complex question of the molecular mechanisms of adrenocortical tumors. The debated topics are exhaustively described, summarizing the background related to the genetic characteristics and regulation of intracellular signaling and transcription factors in the various tumors.

Only few points need to be revised:

-In the paragraph 4 “The pathogenesis and molecular mechanisms in adrenocortical carcinoma (ACC)” , I would suggest implementing the knowledge related to genetic mutations by including that of the IGF II gene  (insulin-like growth factor 2) and also those on PRKAR1A (protein kinase cAMP-dependent regulatory type I alpha), RPL22 (ribosomal protein L22), CCNE1 (cyclin E1), CDK4 (cyclin dependent kinase 4), TERT (telomerase reverse transcriptase) and TERF2 (telomeric repeat binding factor 2) genes.

- The various figures must be mentioned in the text (only figure 1 is mentioned, figures 2,3,4,5,6,7,8 are missing).

- In the figures 2, 3, 5, 6 and 8 I suggest to indicate with different letters (e.g. A and B) the drawings shown. The letters must also be indicated in the figure legend.

- In the figures legend 3 and 8 I suggest mentioning the role of AXIN2.

- Paragraph 2, page 11, on line 11, “ibortezomib” should be replaced by “bortezomib”.

- Paragraph 3, page 11, on line 23 and paragraph 4, page 14, on line 6 “molecular mechanism”  should be replaced by “molecular mechanisms”.

- The year of publication is missing in the following reference numbers:  6, 7, 13, 22, 23, 24, 26, 27, 28, 32, 37, 43, 44, 51, 61, 66, 67, 74, 80, 83.

- The references number 20 and 75 are the same.

- I suggest making a list of genes abbreviations.

Author Response

Reviewer 1

in their review address the issue of the relationship between gene mutations and their regulation of transcription factors, as well as the association of transcription factor complex formation in aldosterone-producing adenoma (APA), cortisol-producing adenoma (CPA) and adrenocortical carcinoma (ACC).

This paper is conceived through an initial introduction in which are described the general characteristics of the adrenocortical tumors classified into adenomas (APA and CPA) and carcinomas (ACC); a second part addresses the pathogenesis and molecular mechanisms of aldosterone overproduction in APA, a third part the pathogenesis and molecular mechanisms of cortisol production in CPA and a fourth part the pathogenesis and molecular mechanisms in ACC.  In conclusion, the authors highlight the main somatic mutations that have been discovered in genes coding for various proteins and resulting in aldosterone or cortisol hypersecretion in APA and CPA, respectively, or in ACC pathogenesis.

The review is well written and updated in dealing with the complex question of the molecular mechanisms of adrenocortical tumors. The debated topics are exhaustively described, summarizing the background related to the genetic characteristics and regulation of intracellular signaling and transcription factors in the various tumors.

Response:

We are grateful to you for the helpful and constructive comments that have enabled us to improve our study. As indicated in the responses that follow, we revised the manuscript based on your instructions. The replies to your specific comments are as follows.

Minor Concerns:

  1. -In the paragraph 4 “The pathogenesis and molecular mechanisms in adrenocortical carcinoma (ACC)” , I would suggest implementing the knowledge related to genetic mutations by including that of the IGF II gene (insulin-like growth factor 2) and also those on PRKAR1A (protein kinase cAMP-dependent regulatory type I alpha), RPL22 (ribosomal protein L22), CCNE1 (cyclin E1), CDK4 (cyclin dependent kinase 4), TERT (telomerase reverse transcriptase) and TERF2 (telomeric repeat binding factor 2) genes.

Response:

Thank you very much for your points. We revised those points. I hope you can confirm it.

  1. - The various figures must be mentioned in the text (only figure 1 is mentioned, figures 2,3,4,5,6,7,8 are missing).

Response:

Thank you very much for your points. We revised those points. I hope you can confirm it.

  1. - In the figures 2, 3, 5, 6 and 8 I suggest to indicate with different letters (e.g. A and B) the drawings shown. The letters must also be indicated in the figure legend.

Response:

Thank you very much for your points. We revised those points. I hope you can confirm it.

  1. - In the figures legend 3 and 8 I suggest mentioning the role of AXIN2.

Response:

Thank you very much for your points. We revised those points. We added content about Axin2 in figure legend 3 and 8. I hope you can confirm it.

  1. - Paragraph 2, page 11, on line 11, “ibortezomib” should be replaced by “bortezomib”.

Response:

Thank you very much for your points. We revised those points. I hope you can confirm it.

  1. - Paragraph 3, page 11, on line 23 and paragraph 4, page 14, on line 6 “molecular mechanism” should be replaced by “molecular mechanisms”.

Response:

Thank you very much for your points. We revised those points. I hope you can confirm it.

  1. - The year of publication is missing in the following reference numbers: 6, 7, 13, 22, 23, 24, 26, 27, 28, 32, 37, 43, 44, 51, 61, 66, 67, 74, 80, 83.

Response:

Thank you very much for your points. We revised those points. I hope you can confirm it.

  1. The references number 20 and 75 are the same.

Response:

Thank you very much for your points. We revised those points. I hope you can confirm it.

  1. - I suggest making a list of genes abbreviations.

Response:

Thank you very much for your points. We revised those points. We added a list of genes abbreviations. I hope you can confirm it.

Reviewer 2 Report

The authors, Shimada, et al., are trying to review the molecular mechanisms of aldosterone-producing adenoma, Cushing’s adenoma, and adrenocortical carcinomas. This review article was well written, and recent evidence were summarized well. I encourage the authors to reconsider some minor revisions.

  1. Page 2, Line 22. “sodium” should be “sodium ion”
  2. Page 3, Line 8. resulting in increase of aldosterone production via cell depolarization or intracellular acidification. I recommend authors to quate the article by Stindl, et al. (Endocrinology. 2015 Dec;156(12):4582-91).
  3. For all figures, to better understand, I recommend the authors to show nucleus in the figure.
  4. Page 5, Line 3 in figure 2 legend. “sodium ion leak” should be “influx of sodium ion”
  5. Page 7, Line 8-9. Authors need to quate a reference in this sentence. Recent review article by Oki, et al. (Endocr J. 2020 Oct 28;67(10):989-995.) would be suitable.
  6. Page 11, Line 37-47. Di Dalmazi, et al. reported that the molecular mechanisms of PRKACA mutated CPA using RNA sequencing data (J Clin Endocrinol Metab. 2020 Dec 1;105(12):dgaa616.). The report must be useful in this review article.
  7. Page 14, Line 16. “Men1” must be “MEN1”.
  8. Page 14, Line 19. The term of “Li-Fraumeni syndrome” must be added, because it must be helpful to make readers understand well.

Author Response

Reviewer 2

The authors, Shimada, et al., are trying to review the molecular mechanisms of aldosterone-producing adenoma, Cushing’s adenoma, and adrenocortical carcinomas. This review article was well written, and recent evidence were summarized well. I encourage the authors to reconsider some minor revisions.

Response:

We are grateful to you for the helpful and constructive comments that have enabled us to improve our study. As indicated in the responses that follows, we revised the manuscript based on your instructions. The replies to your specific comments are as follows.

  1. Page 2, Line 22. “sodium” should be “sodium ion”

Response:

We are grateful to you for the helpful and constructive comments that have enabled us to improve our study. As indicated in the responses that follows, we revised the manuscript based on your instructions. The replies to your specific comments are as follows.

  1. Page 3, Line 8. resulting in increase of aldosterone production via cell depolarization or intracellular acidification. I recommend authors to quate the article by Stindl, et al. (Endocrinology. 2015 Dec;156(12):4582-91).

Response:

Thank you very much for your points. We revised those points. I hope you can confirm it.

  1. For all figures, to better understand, I recommend the authors to show nucleus in the figure.

Response:

Thank you very much for your points. We revised those points. I hope you can confirm it.

  1. Page 5, Line 3 in figure 2 legend. “sodium ion leak” should be “influx of sodium ion”

Response:

Thank you very much for your points. We revised those points. I hope you can confirm it

  1. Page 7, Line 8-9. Authors need to quate a reference in this sentence. Recent review article by Oki, et al. (Endocr J. 2020 Oct 28;67(10):989-995.) would be suitable.

Response:

Thank you very much for your points. We revised those points. I hope you can confirm it

  1. Page 11, Line 37-47. Di Dalmazi, et al. reported that the molecular mechanisms of PRKACA mutated CPA using RNA sequencing data (J Clin Endocrinol Metab. 2020 Dec 1;105(12): dgaa616.). The report must be useful in this review article.

Response:

Thank you very much for your points. We revised those points. I hope you can confirm it

  1. Page 14, Line 16. “Men1” must be “MEN1”.

Response:

Thank you very much for your points. We revised those points. I hope you can confirm it

  1. Page 14, Line 19. The term of “Li-Fraumeni syndrome” must be added, because it must be helpful to make readers understand well.

Response:

Thank you very much for your points. We revised those points. I hope you can confirm it